# Seasonal variation in water use for hygiene in Oromia, Ethiopia, and its implications for trachoma control: An intensive observational study

Katie Greenland[1]*, Alexandra Czerniewska[1], Meseret Guye[2], Demitu Legesse[2], Asanti Ahmed Mume[2], Oumer Shafi Abdurahman[2,3], Muluadam Abraham Aga[2], Hirpha Miecha[4], Gemechu Shumi Bejiga[4], Virginia Sarah[5], Matthew Burton[3], Anna Last[3]

1 Environmental Health Group, Department for Disease Control, London School of Hygiene & Tropical Medicine, London, United Kingdom, 2 The Fred Hollows Foundation, Addis Ababa, Ethiopia, 3 Clinical Research Department, London School of Hygiene & Tropical Medicine, London, United Kingdom, 4 Oromia Regional Health Bureau, Addis Ababa, Ethiopia, 5 The Fred Hollows Foundation, London, United Kingdom

* katie.greenland@lshtm.ac.uk

**Data Availability Statement:** The Oromia Regional Health Bureau Ethics Committee requires that all data sharing requests are reviewed and approved

## Abstract

If facial hygiene practices vary seasonally this could have important implications for the design of interventions for trachoma control. This observational study was conducted to explore seasonal variation in hygiene behaviours in 9 households with at least one child aged 1–9 years-of-age in the West Arsi zone in rural Oromia, Ethiopia. Sixty-one household members were observed intensively over two days in the dry season (January), the rainy season (July) and during the harvest period (October) in 2018. Structured record forms were used to document household water availability and use. Daily water use per capita was very low in all seasons (3.1–4.2 litres). Around one third of water consumed in households in all seasons was associated with body washing. Soap was used during 44 of 677 (6%) of these observed occasions and half of all body washes (n = 340; 50%) included face washing. Overall, 95% of 58 individuals washed their faces at least once between 06:30h and 21:30h in the dry season (21% with soap), compared with 79% in the rainy season (2% with soap) (p = 0.013). Sixty-five percent of householders washed their faces during the harvest observation period (06:30h to 17:30h), none of whom used soap. Twenty-eight percent of 204 children aged 11 and under still had ocular or nasal discharge on their faces after washing. Seventy-three percent of those who washed their faces did so more than once in the dry season, compared with 33% in the rainy season (p<0.001). Face washing occurred throughout the day during the dry season, with a clear peak in the early morning and extra washes in the early evening. Face washing mainly took place in the early morning in the other two seasons. Genuine water scarcity in this area is likely to limit the impact of face washing interventions for trachoma control in the absence of water supply interventions. However, face washing was most common at the time of year when water is the hardest to come by, and seasonal differences in behaviour should be considered in any resulting intervention design.

by them before data can be shared. Data is available to any researcher under reasonable request. To facilitate the data access process please contact ethics@lshtm.ac.uk.

**Funding:** This work was funded by the Wellcome Trust (wellcome.org) through a collaborative award (Grant Number 206275/Z/17/Z) awarded to MB. The funders had no role in study design, data collection and analysis, decision to publish, or preparation of the manuscript.

**Competing interests:** The authors have declared that no competing interests exist.

## Author summary

We conducted a study of household behaviour in rural Oromia, Ethiopia. The study aimed to document facial hygiene practices and any seasonal variation in practices that could be relevant for trachoma control. We observed 61 household members in 9 households with one or more child aged 1–9 years-of-age over two days in three seasons in 2018: the dry season (January), the rainy season (July) and during the harvest period (October). Householders washed their faces more frequently and more often with soap during the dry season than at other times of year. Seasonal differences in behaviour should be considered in the design of interventions to increase face washing with soap in this setting. However as household water availability was limited in all seasons (3.1 to 4.2 litres), genuine water scarcity in this area is likely to limit the impact of face washing interventions for trachoma control in the absence of water supply interventions.

## Introduction

Ethiopia experiences 50% of the world's burden of the blinding infectious disease trachoma, with an estimated 68 million people at risk of disease [1]. Several hygiene practices are believed to be important in limiting the ocular spread of the bacterium *Chlamydia trachomatis*, the causative agent of trachoma [2,3]. Hygiene interventions promoting behaviours such as face and hand washing, as well as the regular washing of clothing and bedding that may act as fomites for transmission are recommended to form a central part of trachoma control [3,4].

Seasonal variations in rainfall play an important role in determining household water availability in rural Ethiopia where 5.7 million individuals rely on surface water sources for their drinking and domestic water needs [5]. In the dry season, when nearby rivers and springs often fail and water collection times lengthen, household water availability can be severely constrained [6]. As water use for hygiene purposes is closely related to total household water availability [7], personal hygiene behaviours such as body washing and the cleaning of clothing and bedding can be restricted when water is scarce. In addition, prioritisation of children's hygiene can also be influenced by agricultural labour demand, with seasonal changes in work load influencing the time available for water collection, childcare and domestic work [8,9].

Given the important influence of seasonality on hygiene practices, it is pertinent to better understand seasonal patterns of water availability and use before designing trachoma control programmes in a chosen setting. However, when conducted, formative research to understand local practices and drivers of behaviour usually takes place during the dry season when remote communities can be most easily accessed. Consequently, resulting interventions often fail to adequately address seasonal barriers that constrain performance of hygiene behaviours.

In 2016, we first conducted formative research on hygiene behaviours in Oromia, Ethiopia [10]. This research was conducted during the dry and rainy seasons and was intended to inform a behaviour change intervention for trachoma control. Counter to our hypotheses and the findings from a similar study conducted elsewhere in the region [6], household water availability was similar in both seasons, despite real water scarcity in the dry season. Furthermore, face washing was more prevalent in the dry season than the rainy season. This earlier study has some limitations. Due to night time travel restrictions, we were reliant on self-reports of evening hygiene routines. In addition, data collection coincided with holidays in both seasons and this may have influenced the documented practices.

Therefore, the current study was conducted to further explore seasonal variation in hygiene behaviours in rural Oromia to inform the development of a face washing intervention for trachoma control.

## Methods

### Ethics and consent

The research protocol was approved by the Ethics Committees of the London School of Hygiene & Tropical Medicine, Ethiopian Federal Ministry of Science and Technology and Oromia Regional Health Bureau. Informed written consent was obtained for each household member 18 years or over. Informed written consent was obtained from parents or guardians on behalf of children under 18 years, and those aged 7 to 17 years also provided assent.

### Study setting

This study took place in two rural *kebeles* (villages) in the Shashemane *woreda* in the West Arsi Zone in Oromia, Ethiopia where trachomatous inflammation–follicular (TF) prevalence is high (TF in children aged 1–9 years old >40%) [11] and where further work including a clinical trial was planned [12].

### Participatory mapping

Separate groups of men and women with at least one child aged 1–9 years old were recruited by health volunteers in each *kebele* and invited to participate in a focus group discussion. The four group discussions aimed to produce a seasonal calendar to inform the timing of seasonal data collection, and aid interpretation of any observed differences in face washing with water and / or soap across the year observed during the seasonal observation study. Participants debated the occurrence of seasonal variation in rainfall, holidays, agricultural activities, water availability, household income and workload and generated a visual representation of variations of each phenomenon over a year. Discussions were audio-recorded and detailed field notes were taken.

### Seasonal household observation

Nine households with at least one child aged 1–9 years old with active trachoma (TF and/or TI–trachomatous inflammation–intense) and with both a pre-school and school-age child at home were purposively invited to participate in the study in order of their detection identified through conjunctival examination of a convenience sample of children. These selection criteria enabled us to study hygiene behaviour among households with children with active trachoma. No households included participants from focus group discussions. Conjunctival examination was performed by an ophthalmic nurse validated in trachoma grading using validated trachoma grading methods described previously [12]. Briefly, the nurses examined the upper tarsal conjunctiva of both eyes using a 2.5× binocular loupe and assigned grades using the WHO Simplified Trachoma Grading System [13]. Conjunctival swabs were collected from each child with active trachoma in the study households and tested for the presence of *Chlamydia trachomatis (Ct)* DNA using a validated *Ct*-qPCR assay [12]. Neither the children who were examined nor their families were directly informed that the children had active trachoma to avoid prompting changes in behaviour during the study. This did not affect the provision of treatment for these children, as mass drug administration with azithromycin was given to the entire community according to national trachoma control policy following the study.

Household members in this cohort were observed intensively over two days on three occasions during 2018: in January (dry season); in July (rainy season); and in October (harvest). These times of year were informed by the participatory mapping and aimed to reflect seasonal variations in rainfall and daily life, factors hypothesised to influence household water availability and body washing practices. Observation sought to capture water use activities in the home throughout the day and evening and took place over 27 hours from the time of waking on Day 1 (approximately 06:30h) until 09:30h the following morning in January and July, temporarily ceasing while the family slept. Observation took place from 06:30h until 17:30h on Day 1 and from 06:30h until 09:30h on Day 2 in October (the harvest season). By observing two consecutive mornings we sought to understand whether early morning body washing rituals—the time of day when we previously observed that washing was most common [10]—occurred in the same way from day to day. Observation in each household took place on the same days of the week in each season to improve comparability from season to season.

Observation was carried out by female Ethiopian field workers who spoke *Afan Oromo*. Data were collected using a structured record form developed in previous studies in Ethiopia and elsewhere and modified following piloting in the current setting. The following information was captured: water availability in the home at the start and end of the observation period (computed by summing the volume of water held in all available vessels i.e. including any rainwater that had been harvested); volume of water collected during the observation period; frequency of body washing, including the body part washed, the person washed and doing the washing, any items used to wash and dry the face, and soap use; and all instances where water was lent to a neighbour or used in the home, including the activity performed (except for drinking, which consumed very small quantities of water at any one time) and an estimate of the volume of water used. Householders returning home were also asked if they had performed any activities involving water at a water point while they were away from the home. To minimise potential reactivity in face washing or soap use, participants were informed that the study aimed to capture seasonal variations in daily life and water use. Particular interest in personal hygiene and face washing was not mentioned. The presence of ocular and/or nasal discharge on children's faces was observed shortly after face washing as soon as the enumerator was close enough to inspect the child's face without arousing suspicion about the purpose of the study. Field workers were specifically trained to make an assessment of the presence of ocular and nasal discharge and followed clear standard operating procedures. In order to assign the presence of ocular discharge, active discharge from the eye must have been present. Simple eyelash crusting was not sufficient to assign the presence of ocular discharge. To assign the presence of nasal discharge, there must have been active discharge from one or both nostrils. Simple crusting around the nose was not sufficient to assign the presence of nasal discharge [12]). Repeat observations in different seasons were carried out by the same field worker whenever possible. Enumerators were trained during a week-long training involving didactic and practical sessions. They did not give any instruction or advice, they positioned themselves discretely and unobtrusively within the family compound and they did not follow family members if they left the home. Female primary caregivers were interviewed following completion of observation on Day 2 to clarify any observed activities.

### Risk factor survey

A risk factor survey capturing data on socioeconomic status, sanitation facilities and reported water availability and use was conducted following the completion of observation in the dry and rainy seasons. This information was collected to ascertain whether actual and perceived water availability differed at different times of year and to help interpret observed data. The

risk factor survey was also carried out in a population-based sample of 247 nearby households in the dry season. Full details of the sampling for this survey can be found elsewhere [12].

## Data analysis

Audio recordings from the four focus group discussions were transcribed and analysed together with field notes so that a unified seasonal calendar could be built. The final calendar reflects opinions that were consistent across the groups on the range and timing of activities.

Paper records from the household observation and surveys underwent quality control checks at the end of each day prior to entry into an Access database. Data were transferred to *Stata 15.1* for data cleaning and analysis. Basic descriptive statistics were performed to provide summaries of observed water availability, water use and body washing behaviour occurring among household members during the 54 observation sessions (Day 1 and Day 2 observation "sessions", i.e. 2 sessions per season in each household). Simple comparisons of means or proportions were made using a repeated samples t-test or Pearson's chi-squared test, respectively. Face washing was defined as any deliberate action taken to wash a face with water only or with water and soap, either on its own, or alongside other body washing. Analysis was restricted to an observation period when inter-seasonal comparisons could be made. *P*-values for comparisons in behaviour across seasons were obtained from random effects logit models, accounting for clustering at the household level.

Total daily water consumption was calculated in two ways:

*Method 1*: (litres of water available in the household at start of day + litres of water collected or received from neighbours during the day)–(litres of water lent to neighbours + litres of water available in household at end of observation period).

*Method 2*: ∑ observed water use during household activities. Computed as the sum of all estimated litres of water used for any purpose (except drinking as described above) during the observation period.

Owing to the shorter observation period at harvest time, Method 2 was included to allow for comparisons across seasons.

## Results

### Profile of households and participants

A cohort of 61 individuals in nine households from two *kebeles* in Shashemane *woreda* (district) were recruited. Each household was observed on the same two consecutive days of the week in three seasons in 2018: January (dry season), July (rainy season) and October (harvest time). Observation took place for a total of 50 hours in each household: from 06:30h until 09:30h the following morning (excepting while the family slept) in the dry and rainy seasons, and from 06:30h until 17:30h on Day 1 and from 06:30h until 09:30h on Day 2 during harvest time. Prevalence of trachomatous inflammation–follicular (TF) in children 1–9 years-of-age was 64% (21/33), of whom 8 (24%) had ocular *Chlamydia trachomatis* infection when examined in the dry season.

Key characteristics of participating households and participants are summarised in Table 1. Respondents had little formal education and only three reported that all school-age children in their household were currently attending school. The predominant water source in the dry season was public taps, with a reported collection time ranging from 20 minutes to 6 hours. Water collection was less time-consuming in the rainy season and some households harvested rainwater. Only two households reported having sufficient water to meet their needs during the dry season, compared with all nine households in the rainy season. All but one household had access to a simple pit latrine, but not all these latrines were observed to be in use. Six

**Table 1. Characteristics of Seasonal Study Households.**

| | Seasonal Study Households (N = 9) | Population-based Sample (N = 247) |
|---|---|---|
| Household size (mean, range) | 6.8* (5–8) | 5.7 (2–12) |
| Education level (primary caregiver) (n, %) | | |
| *No formal schooling* | 5 (55.5%) | 139 (56.5%) |
| *Some primary* | 4 (44.4%) | 83 (33.7%) |
| *Completed primary or higher* | 0 | 24 (9.8%) |
| Radio ownership (n, %) | 5 (55.6%) | 41 (16.7%) |
| Mobile phone ownership (n, %) | 7 (77.8%) | 127 (51.6%) |
| Has (solar) electricity (n, %) | 4 (44.4%) | 31 (12.6%) |
| No. animals owned (mean, range) | 11 (1–30) | 7.0 (0–57) |
| No. types of animals owned (mean, range) | 3.1 (1–8) | 3.1 (0–8) |
| Main water source (dry season) (n, %) | | |
| *Surface water* | 1 (11.1%) | 49 (19.9%) |
| *Public tap* | 8 (88.9%) | 176 (71.5%) |
| *Unprotected well or spring* | 0 | 11 (4.4%) |
| *Piped* | 0 | 8 (3.2%) |
| *Other* | 0 | 2 (0.8%) |
| Water collection time (dry season)** (mean, range) | 245 minutes (20–720 minutes) | 100 minutes (1–480 minutes) |
| Perceived sufficient water to meet needs (dry season) (no. %) | 2 (22.2%) | 49 (19.9%) |
| Perceived sufficient water to meet needs (rainy season) (no. %) | 9 (100%) | 231 (93.9%) |
| Volume water available in home (mean litres, range) | 69.4 (12–140) | 36.8 (0–320) |
| Access to a latrine (n, %) | 8 (88.9%) | 144 (58.5%) |
| Access to a latrine in use*** (n, %) | 5 (55.6%) | 113 (45.7%) |
| Human faeces seen in compound (n, %) | 5 (55.6%) | 72 (29.3%) |
| Soap seen in the home (n, %) | 3 (37.5%) | 87 (35.4%) |

Data for the seasonal households are provided for the dry season to allow comparison with the population-based survey that was conducted at this time.

* Two households had a new baby at the final round of data collection in the harvest season (October 2018).

** Time to collect water is based on self-report and includes time to make a return trip, including any waiting time.

*** Latrine use was assessed using a checklist including observation of faeces or maggots, smell, and other physical features of the latrine.

households did not have any soap in the household at the time of the dry season survey and five did not have soap in the rainy season. The nine households included in the seasonal study had more assets than the wider population, but had broadly similar water, sanitation and hygiene characteristics to the population surveyed during the dry season in 2018 (**Table 1**).

## Seasonal variation in daily routine

Fig 1 displays the seasonal calendar created following discussion with 29 individuals during four gender-separated focus group discussions during the dry season in each of the study *kebeles*. The timing of the three periods of data collection in study households is also shown.

The perception of the year revolves around the agricultural timetable and participants consequently identified seasons based on the agricultural activities performed at that time. Rainfall

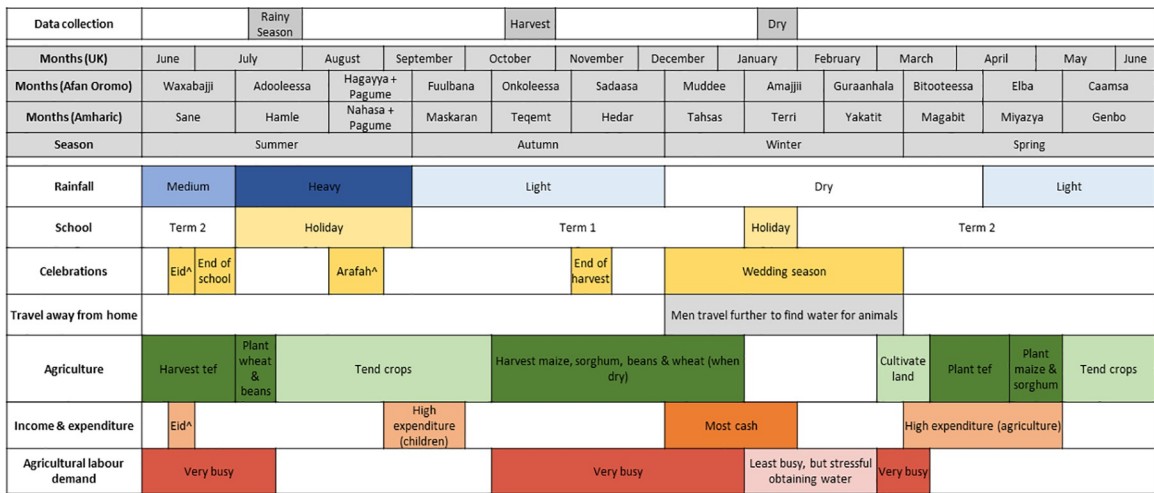

**Fig 1. Seasonal Calendar.**

fluctuates between years and affects crop yields, agricultural labour requirements and ease of water collection. The agricultural cycle creates three main busy periods during the year when the land is cultivated and crops are harvested. Life is quietest at the start of the dry season in December following the harvest. Money and time are most plentiful at this time and the dry season thus coincides with the wedding season. Money is tightest in September (start of the Ethiopian year) following outlay on schooling, as well as certain taxes.

## Seasonal variation in household water availability and use

Table 2 shows household water availability and use during the 27 day-long observations conducted in the dry, rainy and harvest seasons on Day 1.

## Water collection

Households had water available for use first thing in the morning on all but two observed occasions. Water collection was observed infrequently: water was not collected at all during half (14 of 27) of the day-long observation sessions, and three of the nine households were not observed to collect water from any source in any season. Twenty-three (72%) of 32 successful water collection trips were performed on foot and the remainder involved use of a donkey or hand cart. Water was collected by both children (59% of trips) and adult females (41% of trips), usually as a discrete trip rather than an add-on to other activities. Water was obtained predominantly from a river and occasionally from public taps or a pond during the dry season, from rivers and ponds in the rainy season and from rivers only during the harvest time. Not all attempts to collect water in the dry season were successful owing to low water pressure or non-functional standpipes. Water collection was reported to take considerably longer during the dry season than during the rainy season (Table 1), but observation revealed water collection took a similar time (when successfully collected) in all seasons (**Table 2**). Water was observed to be lent between households, but almost exclusively during the dry season. Rainwater was harvested in small vessels during the rainy season but was not transferred to larger containers for storage. As rainwater harvesting substituted rather than supplemented water collection from other sources, less water was collected on average during the rainy season than in the dry and harvest seasons (**Table 2**).

**Table 2. Seasonal variation in household water availability and use during the Day 1 Observation Period.**

| | Dry Season | Rainy Season | Harvest Time |
|---|---|---|---|
| **Length of Day 1 observation period** | **06:30h – 21:30h** | **06:30h – 21:30h** | **06:30h – 17:30h** |
| **Water availability (mean, range)** | | | |
| Litres water in home first thing in morning (06:30h) (A) | 69.4 (12–140) | 53.1 (4–130) | 39.2 (10–78) |
| No. trips to collect water during observation period | 0.9 (0–3) | 0.8 (0–3) | 0.8 (0–3) * |
| Time taken to collect water** | 49 mins (23 mins-1hr48mins) | 53 mins (8 mins-1hr44mins) | 45 mins (15mins-1hr20mins) |
| Litres water collected during day (B) | 17.7 (0–80) | 9.0 (0–45) | 16.1 (0–50) |
| Litres water in home in evening (C) | 36.8 (0–95) | 32.7 (0–102) | 32.8 (19–55)* |
| **Total observed household water use in litres between 06:30h and 17:30h (mean litres, % of total)** | | | |
| Body wash | 11.0 (31.7%) | 7.7 (39.0%) | 6.1 (31.4%) |
| Cooking | 4.1 (11.8%) | 3.7 (19.0%) | 5.1 (26.4%) |
| Dishes | 4.3 (12.5%) | 3.6 (18.2%) | 3.4 (17.5%) |
| Coffee-related | 2.4 (6.8%) | 2.9 (14.5%) | 2.7 (13.8%) |
| Laundry | 3.9 (11.3%) | 0.3 (1.7%) | 0.8 (4.0%) |
| Used to moisten floor | 0.7 (2.1%) | 0.2 (1.1%) | 0.1 (0.3%) |
| Given to animals | 1.2 (3.3%) | 0.1 (0.3%) | 0.6 (2.9%) |
| Other | 0.03 (0.1%) | 1.0 (5.1%) | 0.2 (1.1%) |
| Lent to a neighbour (D) | 7.1 (20.4%) | 0.2 (1.1%) | 0.5 (2.6%) |
| **Total daily water consumption (mean litres, range)** | | | |
| Estimated household water use during total observation period*** <br> *Method 1: (A+B)–(C+D)* | 42.6 (16–85) | 29.4 (16–48) | 22.6 (4–54)* |
| Estimated household water use during total observation period*** <br> *Method 2: ∑ observed water use* | 38.0 (13.2–79.9) | 22.9 (12.5–37.0) | 18.9 (2.2–44.3)* |
| Estimated household water use for comparable period (06:30h - 17:30h) <br> *Method 2: ∑ observed water use* | 27.7 (6.5–64.3) | 19.5 (10.0–37.0) | 18.9 (2.2–44.3) |
| Estimated water use per capita for comparable period (06:30h - 17:30h) <br> *Method 2: ∑ observed water use* | 4.2 (0.9–8.0) | 3.1 (1.4–6.2) | 3.1 (0.3–7.4) |

* Not directly comparable with the other seasons due to the shorter observation period at harvest time.

** Time taken to collect water is based on observations of occasions when someone returned with water. Unsuccessful trips, when water collection was attempted but not achieved e.g. due to a broken tap, are not included.

*** Total observation period is 06:30h – 21:30h in the dry and rainy season and 06:30h – 17:30h in the harvest season.

Total household water use in litres does not include water used for drinking as such small quantities of water were consumed at any one time.

Method 2 was used to estimate mean household water use and water use per capita in the total daily water consumption section so that comparisons can be made across the seasons.

**Water use.** Table 2 also shows how households used water at home between 06:30h and 17:30h on 1114 occasions during the 27 Day 1 observation sessions. Body washing–which reportedly almost always occurs at home in all seasons—consumed the greatest proportion of available water in all seasons (31–39%), followed by cooking (12–26%), washing dishes (13–18%), and preparing coffee (7–15%) (Table 2). Seasonal study and population-based survey respondents both reported that they had sufficient water to meet their needs in the rainy season but not in the dry season (Table 1, $p<0.001$ in both studies). However, mean observed household water consumption was significantly higher in the dry season than during a comparable period of time in the rainy season (28 litres vs. 20 litres, $p = 0.02$) or harvest period (28 litres vs. 19 litres, $p = 0.02$) (Table 2). Water use per capita was low in all seasons (3.1 litres rainy and harvest seasons, 4.2 litres dry season). More water was used in the dry season during both the day and in the evening than in the rainy season. Including water used in the evenings in the dry and rainy seasons increases water use per capita to 5.8 litres and 3.6 litres, respectively.

According to population-based survey participants, activities consuming large amounts of water such as clothes washing take place at home the majority of the time (90% rainy season vs. 65% dry season, $p<0.001$), although they are performed less frequently in the dry season (91% of 246 survey participants reported washing children's clothing one or more times a week in the rainy season and 71% reported doing so in the dry season). Laundry was infrequently observed at home (8 times in total during 585 hours of observation on different days) and clothing was only taken to the river to wash by one household in the dry and harvest seasons and not at all in the rainy season. Bedding was not washed during any observation session.

## Seasonal variation in body washing

Table 3 shows 552 observed body washes by season, body part and soap use for a comparable time period from 06:30h to 17:30h on Day 1 and from 06:30h to 09:30h on Day 2 during the 54 observation sessions. Forty-two (8%) of these body washes involved soap. Overall, 220 body washes (10% with soap) occurred in the dry season, 187 (6% with soap) in the rainy season and 145 (6% with soap) at harvest time. Half of all body washes (291 of 552) included face washing, but only 27 (9%) used soap. In one household in the dry season soap was purchased during the study and all children in the household were bathed with soap (Table 3). A further 125 body washes (2% with soap) occurred between 17:30h and bedtime (around 21:30h) on Day 1 in the dry and rainy seasons. Faces were washed during 47 (38%) of these washes (35 in the dry season, 12 rainy season), but soap was only used twice. As shown in Fig 2, the additional face washing in the evening in the dry season corresponded to a clear peak in washing at this time that was not observed in the rainy season.

## Seasonal variation in face washing

Table 4 shows the proportion of household members observed to wash their face at least once over the course of the Day 1 observation period in each season. Face washing was more frequent, involved more household members, and more often used soap during the dry season than at other times of year. Overall, 95% of 58 individuals washed their faces at least once in the dry season (21% with soap), compared with 79% in the rainy season (2% with soap) ($p<0.013$). Sixty-five percent of householders had washed their faces by 1730h during the harvest observations, none of whom used soap. Among those who washed their faces in each season, 73% (40/55) did so more than once in the dry season, compared with 33% (15/45) in the rainy season ($p<0.001$).

The majority (80%) of 50 face washes among pre-school children aged 0–2 years were done by an older sibling or caregiver, but children aged 3–6 years more often washed their own

**Table 3. Seasonal variation in observed body washing during comparable observation periods on Day 1 and Day 2.**

| Body part washed | Dry Season (n, %) | | Rainy Season (n, %) | | Harvest Time (n, %) | |
|---|---|---|---|---|---|---|
| | Water only | Water & Soap | Water only | Water & Soap | Water only | Water & Soap |
| Hand(s) only | 80 (98.8%) | 1 (1.2%) | 88 (100.0%) | 0 | 57 (100%) | 0 |
| Face and hand(s) | 85 (95.5%) | 4 (4.5%) | 65 (94.2%) | 4 (5.8%) | 65 (97.0%) | 2 (3.0%) |
| Face, hand(s) and other | 23 (69.7%) | 10 (30.3%) | 16 (76.2%) | 5 (23.8%) | 10 (83.3%) | 2 (16.7%) |
| Hand(s) and / or other | 8 (100%) | 0 | 6 (100%) | 0 | 3 (75.0%) | 1 (25.0%) |
| Full bath | 2 (22.2%) | 7 (77.8%) | 1 (33.3%) | 2 (66.7%) | 1 (20.0%) | 4 (80.0%) |
| Total | 198 (90.0%) | 22 (10.0%) | 176 (94.1%) | 11 (5.9%) | 136 (93.8%) | 9 (6.2%) |

Table shows row percentages for each season.

Data reported are for a comparable time period from 06:30h to 17:30h on Day 1 and from 06:30h to 09:30h on Day 2.

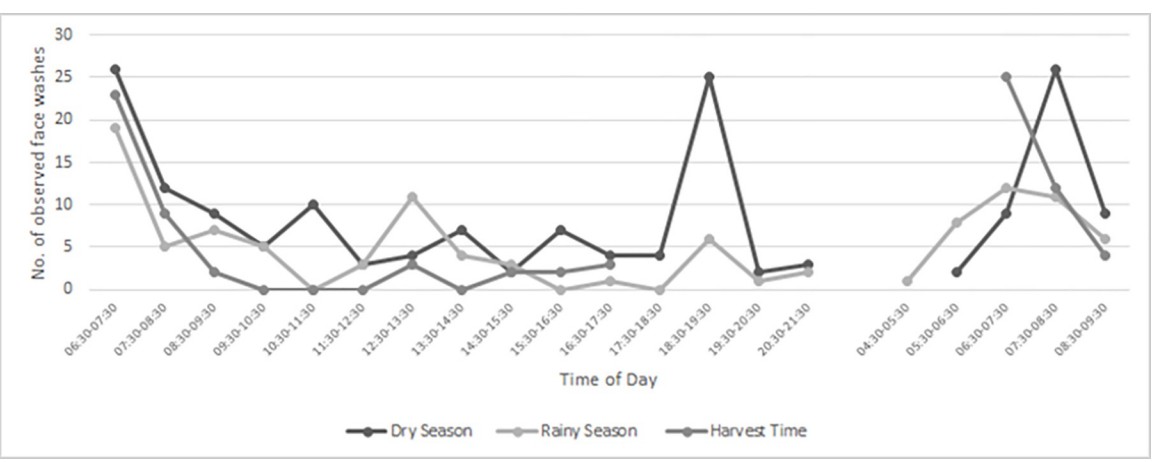

**Fig 2. Timing of observed face washes by season.**

faces (68/88 washes, 77%). Soap use was rare irrespective of who washed a preschool child's face (11/61 washes by a caregiver vs. 4/77 self-washes, $p = 0.026$). Faces were left to air dry after washing on almost all occasions (327/338, 97%). Overall, 28% of children aged 11 and under had ocular and/or nasal discharge on their faces when observed post-face wash (N = 204). Oculo-nasal discharge on faces following face washing varied seasonally: 25/104 (24%) dry season; 7/49 (14%) rainy season; and 25/51 (49%) at harvest time ($p = 0.022$).

Faces were also reportedly washed at the water source by 15 individuals–mainly children collecting water–in the dry season, three in the rainy season and five at harvest time.

Fig 2 shows the times of day that faces were washed during the Day 1 and Day 2 observation sessions. Face washing occurred throughout the day during the dry season, with a clear peak in the early morning and early evening. Face washing mainly took place in the early morning in the other two seasons and no evening peak was observed in the rainy season. The frequency

**Table 4. Seasonal variation in proportion of household members washing faces over the course of the Day 1 observation perio.**

| | N | Dry Season (n, %) | | | Rainy Season (n, %) | | | Harvest Time (n, %) | | |
|---|---|---|---|---|---|---|---|---|---|---|
| | | Face not washed | Face washed with water | Face washed with soap | Face not washed | Face washed with water | Face washed with soap | Face not washed | Face washed with water | Face washed with soap |
| **Father** | 8 | 1 (12.5%) | 6 (75.0%) | 1 (12.5%) | 1 (12.5%) | 7 (87.5%) | 0 | 4 (57.1%) | 3 (42.9%) | 0 |
| **Mother** | 9 | 1 (11.1%) | 5 (55.6%) | 3 (33.3%) | 0 | 8 (88.9%) | 1 (11.1%) | 3 (37.5%) | 5 (62.5%) | 0 |
| **School-age child (12–18 yrs)** | 3 | 0 | 2 (66.7%) | 1 (33.3%) | 1 (33.3%) | 2 (66.7%) | 0 | 2 (66.7%) | 1 (33.3%) | 0 |
| **School-age child (7–11 yrs)** | 12 | 0 | 10 (83.3%) | 2 (16.7%) | 4 (33.3%) | 8 (66.7%) | 0 | 3 (23.1%) | 10 (76.9%) | 0 |
| **Pre-school child (3–6 yrs)** | 17 | 1 (5.9%) | 12 (70.6%) | 4 (23.5%) | 3 (17.7%) | 14 (82.4%) | 0 | 3 (18.8%) | 13 (81.3%) | 0 |
| **Pre-school child (0–2 yrs)** | 9 | 0 | 8 (88.9%) | 1 (11.1%) | 3 (33.3%) | 6 (66.7%) | 0 | 5 (50.0%) | 5 (50.0%) | 0 |
| **Total** | 58 | 3 (5.2%) | 43 (74.1%) | 12 (20.7%) | 12 (20.7%) | 45 (77.6%) | 1 (1.7%) | 20 (35.1%) | 37 (64.9%) | 0 |

Table shows row percentages.

Denominators vary slightly during the harvest season as follows: two individuals (1 father and 1 mother) from different households were absent during the harvest time observation, 1 child turned 7 and was moved up an age category, and one new baby was observed.

Data reported are for a comparable time period from 06:30h to 17:30h on Day 1 in all seasons.

and timing of face washing on the second morning of observation was similar to Day 1, although the overnight stay allowed additional prayer washes occurring in the very early morning to be captured on Day 2.

## Discussion

We conducted extended household observations on two consecutive days at three times of year to explore how seasonality may affect household water availability and the performance of hygiene behaviours in a rural area near Shashemene in Oromia, Ethiopia. The long observation periods combined with overnight homestays increased our ability to document actual practices of relevance to trachoma control to a level of detail hitherto undescribed in the literature.

Per capita water use was very low in all seasons, but despite this we observed seasonal differences in household water availability and use, with more frequent body washing and associated water consumption in the dry season than in the rainy season or during harvest time. Face washing appeared to be more common in the dry season than at other times of year (although it did not always remove ocular and nasal discharge). Soap use during body and face washing was low in all seasons, but was slightly more common in the dry season. This increase can be largely explained by reactive bathing of children with soap in one household, but may also be partly related to soap availability: although many households did not have soap in any season, participants reported that it is difficult to purchase soap when money is tight during the rainy season and in the lead up to harvest, so it is possible soap is used more freely in the dry season.

Water collection in the dry season took a similar time to the other seasons, despite participants reporting and perception that it takes far longer in the dry season. This difference could simply be because we recorded the length of successful water collection trips only, but several trips to collect water in the dry season were unsuccessful due to issues with standpipes, which took considerable amounts of time and could have caused people to overestimate water collection times (which vary from source to source and day to day and are thus hard to estimate accurately). The finding that households consumed more water at home during the dry season than in the other two seasons is surprising given that the study population perceived water to be scarce and insufficient to meet their needs at this time. The data suggest that water is indeed scarce: water was shared between households in the dry season but not at other times of year, survey respondents consistently reported longer water collection times and several attempts to collect water in the dry season were unsuccessful (which may further entrench perceptions of water scarcity). Our observation that households used more water at home in the dry season is in keeping with data we have collected previously in a site over 400 km away (10), but this remains in stark contrast with findings from a study conducted elsewhere in Oromia (6). Intuitively, in line with the findings of Tucker *et al* (6), we would have expected household water availability and consumption to be highest during the rainy season.

If water is truly harder to come by during the dry season, why did householders have and use more water in their homes at this time? There may be some observer effect, whereby participants' behaviour changed simply as a result of being observed, however our data suggest several possible reasons for this phenomenon. Even though water collection is more laborious and water sources dry up and change over the course of the dry season [14], at the time of the dry season study water was still available. It may also be that perceived and actual scarcity increases the focus on water collection, and coupled with the lighter agricultural workload at this time of year, people have comparatively more time to collect water, so genuine water scarcity does not translate to less water at the household level. As people spend more time at home in this season, they also have more opportunity and need to use water. Due to the small

quantities consumed at any one time we did not record water used for drinking, but it is also possible that more water was used for drinking during the dry season. Our study suggests that some household activities involving water use in the dry season do not occur at other times of year (lending water, watering animals, moistening dusty mud floors). People also washed their faces more frequently at this time, both at home and at the water point, reportedly because the dry season is dusty and face washing is refreshing. Another potential driver of face washing at this time could be the type of food consumed: a shift in diet from potatoes to *injera* pancakes eaten with spices and sauces was observed to prompt face and hand washing (after the meal). Whether faces were washed thoroughly enough to remove ocular and nasal discharge is another matter, as we observed that oculo-nasal discharge was not always removed by washing, with some possible seasonal variation. As enumerators recorded this surreptitiously and could not always closely observe children's faces after washing, these results should be considered indicative of inadequate washing only. We have also observed incomplete removal of oculo-nasal discharge following face washing with water in another study we have conducted in the same setting [15].

Although it is noteworthy that households used more water during the dry season, this should not detract from the fact that the quantity used falls far short of the minimum of 15 L per capita recommended by SPHERE for basic survival-level water requirements [16]. Estimates of household water consumption based on observed water use were in line with those based on water availability, which suggests that the assessment of household and per capita water use was quite accurate. Activities that consume the most water (clothes washing and full bathing) reportedly occur primarily at home, but these activities occur infrequently (even in the rainy season) and this is likely why they were almost never observed during the one and a half days of observation in each season. Even if these activities were carried out on days when households were not observed, the lack of observed bathing and clothes washing in this study correspond with norms of infrequent bathing and laundry reported by participants and documented in our previous study [10]. Households were observed on the same days in all seasons so differences between behaviour across the seasons are likely to be real. However, water collection and consequent consumption in the rainy season is likely to be higher on days when laundry and full body bathing take place.

## What are the implications of these findings for the design of a face washing intervention in this region?

First, it should be noted that in the absence of water supply interventions, the potential for behaviour change is likely to be limited by low household water availability.

This study observed that faces of pre-school children are not washed in a way that removes ocular and nasal discharge. Although inter-observer reliability was not assessed in this study, observers were trained and used clear and simple standard operating procedures to define the presence of ocular and/or nasal discharge throughout the study. A potential intervention should therefore consider emphasising the importance of cleaning around the eyes and nose. As children as young as 3 years-of-age tended to wash their own faces, it would also be worth encouraging caregivers to either wash pre-school children's faces directly or supervise face washing. Our study also identified important seasonal barriers to face washing that should be considered in the design of a face washing intervention in this region, the most important of which is the physical lack of soap in households. It is well-recognised that the rains and time leading up to harvest are associated with the greatest poverty [9,17] and people are likely to be more concerned with other issues affecting their lives than lack of soap for face washing. That said, cost-effective alternatives to bar soap, such as soapy water could be promoted at the start

of "difficult" times of year. Despite real water constraints in the dusty dry season, water is available in households in slightly greater quantities in the dry season and face washing takes place more frequently, despite reports that it is not possible to wash due to the lack of water. As part of wider activities to conserve water such as investing in handwashing stations that conserve water, an intervention could address perceptions of water availability or the amount of water needed to wash faces to encourage face washing in the dry season. Face washing at other times of year could be encouraged by sharing statistics on face washing to show what they are doing in the dry season when water is most precious, through water prioritisation activities and through discussion about work burdens and the lack of time reported by women during times of peak agricultural demand. The early dry season is often a time of greater security and wealth. Reinforcement events should be held at strategic times of year that coincide with observed seasonal barriers.

This study is not without its limitations and it is important to consider how these limitations affect interpretation of the study's findings. Whilst these nine households had many similarities to the wider population in this region with regards to WASH conditions, given the small sample size and relatively high level of asset ownership in the study population we recommend caution with the generalisability of these findings to populations in other regions. Furthermore, as Islam is the predominant religion in the study area and all households included in this study were Muslim, the findings may not be generalisable to communities elsewhere in Ethiopia, where ritual ablution practices are not followed. Although there may be some observer effect seen during the first time point of observation (in the dry season), the duration of structured observation affects the extent to which people behave as normal in a structured observation study ('reactivity') [18,19]. In this study, households were observed for a day and a half at each time point, which is a relatively long period of observation in a study of this kind. We cannot fully discount the possibility of observer bias in the noted increase in frequency of observed body washing in the dry season, but gain reassurance from the fact that family members performed similar hygiene behaviours on two consecutive mornings, with one clear exception being the family who purchased soap during the dry season study to bathe all their children, presumably in reaction to the presence of the study team. Families knew that the study was connected to trachoma but were not aware that their children had active trachoma (see methods) so it is possible that they could have modified their behaviour to increase practice of known trachoma prevention behaviours such as face washing. However, we were conscious of this potential bias and took care to avoid talking about hygiene behaviours when providing information about the study to mitigate against this. Furthermore, if people did have such knowledge about the purpose of the study, it should have been higher on the second and third visits following interview and risk factor survey completion, but despite this we observed more frequent face washing and soap use during the first visit in the dry season. Finally, the statistical comparisons should be interpreted within the context of this small study, which was not powered specifically for this, but to gather rich observational data to better understand seasonal hygiene practices in this trachoma-endemic region. This is crucial in the development of an appropriate household-level hygiene intervention to be tested in a large cluster randomised controlled trial.

## Conclusions

We found seasonal variations in household water availability and hygiene practices that may be relevant to trachoma control. Our study suggests that face washing varies seasonally in West Arsi in the Oromia region of Ethiopia. Face washing appeared to be more frequent during the dry season than at other times of year. This study has also identified several seasonal

barriers to face washing and soap use that should be addressed to improve the success of any face washing interventions developed in this region.

## Acknowledgments

We are immensely grateful to the support of the *kebeles* where this work was conducted, in particular, the participants who allowed our researchers into their homes for intensive observation with overnight stays. We also wish to thank staff at the Fred Hollows Foundation Ethiopia for assistance with the logistics for the field work, particularly our dedicated drivers (Teka Ashagrie, Kibreab Abino and Fitsum Shappa).

## Author Contributions

**Conceptualization:** Katie Greenland, Virginia Sarah, Matthew Burton, Anna Last.

**Data curation:** Katie Greenland, Alexandra Czerniewska.

**Formal analysis:** Katie Greenland, Alexandra Czerniewska.

**Funding acquisition:** Katie Greenland, Virginia Sarah, Matthew Burton, Anna Last.

**Investigation:** Alexandra Czerniewska, Meseret Guye, Demitu Legesse, Asanti Ahmed Mume.

**Methodology:** Katie Greenland, Alexandra Czerniewska.

**Project administration:** Alexandra Czerniewska, Oumer Shafi Abdurahman, Muluadam Abraham Aga.

**Resources:** Oumer Shafi Abdurahman.

**Supervision:** Katie Greenland, Oumer Shafi Abdurahman, Muluadam Abraham Aga.

**Visualization:** Katie Greenland.

**Writing – original draft:** Katie Greenland.

**Writing – review & editing:** Katie Greenland, Alexandra Czerniewska, Meseret Guye, Demitu Legesse, Asanti Ahmed Mume, Oumer Shafi Abdurahman, Muluadam Abraham Aga, Hirpha Miecha, Gemechu Shumi Bejiga, Virginia Sarah, Matthew Burton, Anna Last.

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
