## [Decision Letter · Decision Letter 0]

27 Aug 2021

Dear Dr Greenland,

Thank you very much for submitting your manuscript "Seasonal variation in water use for hygiene in Oromia, Ethiopia, and its implications for trachoma control: an intensive observational study" for consideration at PLOS Neglected Tropical Diseases. As with all papers reviewed by the journal, your manuscript was reviewed by members of the editorial board and by several independent reviewers. In light of the reviews (below this email), we would like to invite the resubmission of a significantly-revised version that takes into account the reviewers' comments. 

The manuscript reports the results of an observational study, of a small number of households in a trachoma endemic community, on water use in different seasons. Please address the issues raised by the reviewers, particularly those concerning the study's methods, and the possibility of biases in the interpretation of the data. Apparently the characteristics of the selected households and the sample of the general population (table 1) seem quite different, as noted by one of the reviewers.

We cannot make any decision about publication until we have seen the revised manuscript and your response to the reviewers' comments. Your revised manuscript is also likely to be sent to reviewers for further evaluation.

Sincerely,

Expedito J A Luna, MD

Guest Editor

Guilherme Werneck

Deputy Editor

The manuscript reports the results of an observational study, of a small number of households in a trachoma endemic community, on water use in different seasons. Please address the issues raised by the reviewers, particularly those concerning the study's methods, and the possibility of biases in the interpretation of the data. Apparently the characteristics of the selected households and the sample of the general population (table 1) seem quite different, as noted by one of the reviewers.

Reviewer's Responses to Questions

**Key Review Criteria Required for Acceptance?**

**Methods**

-Are the objectives of the study clearly articulated with a clear testable hypothesis stated?

-Is the study design appropriate to address the stated objectives?

-Is the population clearly described and appropriate for the hypothesis being tested?

-Is the sample size sufficient to ensure adequate power to address the hypothesis being tested?

-Were correct statistical analysis used to support conclusions?

-Are there concerns about ethical or regulatory requirements being met?

Reviewer #1: There are several issues with the methods, as detailed in the comments. The sample size determination was not provided, the statistical analyses were not detailed sufficiently, and the population not described sufficiently. The most serious issues are around potential biases in these data, which are not discussed.

Reviewer #2: Objectives of the study were well articulated and the researchers proposed an appropriate study design. Methods are adequately described. Although they included a low number of households for the study and the authors recognized that this might be a limitation, the findings of the study are important to contribute to understand better the challenges for trachoma elimination. The authors used adequate analysis methods to support the conclusions. Authors might review and consider to adjust the following in the Methods sections:

1. Line 84: TF stands for trachomatous inflammation-follicular not for follicular conjunctivitis.

2. Definition of "face washing practices" should be clarified for the purposes of the study. 

3. Line 95: TI stands for Trachomatous inflammation– intense.

Reviewer #3: (No Response)

**Results**

-Does the analysis presented match the analysis plan?

-Are the results clearly and completely presented?

-Are the figures (Tables, Images) of sufficient quality for clarity?

Reviewer #1: The results need clarification, especially Tables 3 and 4. Figure 1 can be deleted as described in the detailed comments. The analyses plan is not sufficiently detailed nor is the issue of clustering at household level addressed.

Reviewer #2: Results are very well presented, clear, and matched the analysis plan.

Please consider the following:

1. Line 169: TF stands for trachomatous inflammation-follicular not for follicular conjunctivitis.

2. Line 181: authors compared the characteristics of the 9 households studies to the wider population studied during the dry season in a survey in 2018. For this, authors listed the characteristics of both in table 1. It is not very clear that the characteristics are similar in both. For example, for characteristics such as radio ownership, mobile ownership, and having solar electricity, the 9 studied Households have much higher proportions than the population-based sample. Same happened with other characteristics (main water sources, water collection time, volume of water available in home, access to a latrine). It would be ideal if authors can explain a little bit why they arrived to the conclusion that there are no differences.

3. Line 391: it seems that authors missed to include the number of the reference.

Reviewer #3: (No Response)

**Conclusions**

-Are the conclusions supported by the data presented?

-Are the limitations of analysis clearly described?

-Do the authors discuss how these data can be helpful to advance our understanding of the topic under study?

-Is public health relevance addressed?

Reviewer #1: There are several more limitations than the authors described, some with major impacts on the conclusions. Until the authors address concerns about the biases, it is not clear that some of the conclusions are valid

Reviewer #2: Conclusions are very well supported by the data presented by authors. Limitations were cleared described. Conclusions of this study are important to better understand challenges in making progress towards the elimination of trachoma as a public health problem. The international discussion about the need to better understand how to improve facial cleanliness can be enriched with the findings in this study. Understanding the practices in specific contexts, can help to tailor interventions to accelerate efforts toward the elimination goals.

Reviewer #3: (No Response)

**Editorial and Data Presentation Modifications?**

Reviewer #1: major modifications are described below

Reviewer #2: Minor Revision

Reviewer #3: Methods:

Study setting 

Page 5 line 84 – I would suggest including the name of the country and also to correct – follicular conjunctivitis” (TF) prevalence is high (TF1-9>40%) - “trachomatous inflammation – follicular (TF) prevalence is high (TF in children aged 1-9 years old >40%)”

Please correct throughout the manuscript “trachomatous inflammation – follicular (TF)”- Results page 9 line 169 and “trachomatous inflammation – intense (TI)”- page 5 line 96.

Data analysis

Page 8 line 152- include one parenthesis at the beginning of the explanation 

Page8 line 155 – this is the first time that this symbol “∑ “appears – I suggest including an explanation how the sum of the amount of water was performed.

Results: 

Supplementary Table1 

Many of the information already are described in the text. I think there is no need including the Supplementary Table1 in the manuscript. The information that is not ,can be included in the results section.

Table 2 page 12 

Please explain better the difference between “Estimated household water use during observation period” and “Estimated household water use for comparable period (06:30 - 17:30)”.

I would suggest writing the information more clearly for better understanding “Estimated household water use during total observation period 06:30h-21:30h”.

Table 3 

Lines 279 and 280 - it is written “ Data reported are for a comparable time period from 06:30 to 21:30 on Day 1 and from 0630 to 0930 on Day 2 for the dry and rainy seasons, and for a shorter observation session from 06:30h….” but at lines 271 to 274 says “Removing 272 washes that took place between 530pm and 930pm on Day 1 and before 6:30am on Day 2 from the 273 dataset removes 125 body washes, including 47 face washes (35 in the dry season (2 with soap), 274 in the rainy season (0 with soap) from the dataset”. Please explain the reason for removing the information of washes that took place between 17:30h and 21:30h from the analysis. There were removed from the table 3?

Table 4 

Include in the table that it is missing two information’s one father and one mother at the harvest time.

Page 16 lines 289 and 290 – Please explain how these calculations were done. “Among those who washed theirs faces” (dry season – 55, plus rainy season 46, plus harvest time 37 = 138); dry season 55/138*100=40%; rainy season 46/138*100= 33%. My calculations do not result the number that it is written.

Discussion

Page 22 line 391 – “As part of wider activities to conserve water such as investing in innovative tools [ref]”…. I belie that the authors intended to include a reference there, but it is missing. 

References

Some references should be corrected, because are not Vancouver style. Please make necessary corrections of the references numbers 1, 4, 5 and 8.

**Summary and General Comments**

Reviewer #1: Understanding water usage and washing practices in communities with trachoma is vital to designing intervention packages, and this manuscript provides data on a limited sample in Oromia, Ethiopia. These intense observational studies are difficult to do, and we do not expect the sample sizes typically seen in epidemiological studies. However, to be worthwhile for programs, as the authors state is the intent of the work, there must be some generalizability in a larger sense, and assurances that there are limited biases in the observations. These concerns are especially relevant when there are observations on just 9 households over just one day and a half in 3 different seasons. The authors must provide more information to the readers to assuage the concerns. Can the authors please provide the following: 

Selection of the sample

1. How was the sample size determined-both in number of sub-kebeles, number of households and number of days to represent the seasons/period? 

2. What were the populations sizes of these sub-kebeles? 

3. Line 83: It is said the sub-kebeles were “purposefully selected”. What were the criteria for selection? Out of how many sub-kebeles were these chosen? 

4. Line 97. The households were also purposefully selected, but we only get a list of partial criteria, later in the manuscript: The houses had to have one preschool and one school age child, and one child had to have trachoma. We also learn that 21 children had trachoma-so how were the 9 households selected from among these households? Why did the households have to have trachoma? 

Bias: The authors have a very limited discussion of the potential for bias in these findings, but there are reasons to be concerned. Can the authors please provide information on the following: 

5. It appears the trachoma survey was carried out prior to the observations, which means the families were aware of the status of their children. Given the emphasis on F ad E in the Ethiopian trachoma program, is it possible that families altered behavior for observers because they knew their child was affected? The authors need to discuss this in the manuscript and address this issue if they somehow do not feel it was a factor and why not. 

6. Line 104. Were all the seasons observed in the same order for each household? That is, the first set of observations for all was dry season, then rainy season, then harvest season? If so, this is a serious limitation, as it is very possible that families increased washing practices and increased water availability for the first few days of observation, wishing to appear in good light to the observer, but then in subsequent seasons with more days, family practices settled down to normal routine. One way to address this is to have the first days of observation vary by season in different households and see if there is a temporal trend across seasons. Had different households started observations in different seasons one could parse out the effect of first observation from season.( Another way is to increase the number of days of observation in each season and determine if there is a change over the days; this was not done but was an option). As it is, the results show more water and more washing in the first set of observations which coincides with the dry season. The authors conclude there is more water usage and washing in dry season, rather than a Hawthorne effect of the timing of the initial observations. If there is bias, one would expect washing to decline over subsequent observations, which it does. The authors need a fuller discussion of this issue as this reviewer is unconvinced that there is more water in the household and more washing done in the dry season from these data. 

7.. The focus groups in these two sub kebeles were made aware of the study interest in water and washing. Were any of the households in the study part of the focus group? 

 a. If so, this needs to be explicit. The discussion needs to reflect this major limitation , that washing as a focus of the research was known to some of the study participants. 

 b. If not, how did the authors ensure that members of the focus group did not reveal the subject matter to the rest of the community. 

8. The authors state that the observers were unobtrusive, yet state that they observed nasal and ocular discharge not cleaned from the children’s faces after washing. Can the authors please address the following: 

a. How were the observers standardized? What was the definition used and what was the agreement between observers? 

b. It is not possible to ascertain ocular discharge without close observation. It is difficult to imagine how facial cleanliness was observed , right after washing a child’s face, unobtrusively. This would have indicated special interest in face washing in particular and again made participants aware of the study purpose. Can the authors please elaborate on this concern in the discussion and methods? 

9. Considering the above, how can the authors state the Hawthorne effect was not a problem (line 406), especially considering the temporal (rather than seasonal) changes discussed above 

10. Page 150: Significance testing is exceedingly difficult to do in these types of studies, because there are only 9 families, and data within each family are clearly correlated, as are observations over time in the same family/person. Longitudinal data analyses using repeated measures is reasonable, but some consideration must be given to clustering within families. It is difficult to see how a simple chi square could be used as it requires assumptions of independence and clustering within households is an issue. When comparisons with another population-based survey are used, how are the p values derived? Are they adjusted for household clustering? Please clarify how the p values were derived for testing within the 9-household survey. 

11. The manuscript has other areas that need more clarity for the reader. One is in the use of sessions and observation periods. In some places, there is a reference to 54 sessions (line 148) and others it is 27 sessions (line 229). In the methods, could the terms and numbers be clarified-did the authors mean that the nine households had six observation periods-one long first day, broken into two sessions plus two distinct periods for the harvest season? (Line 108)It is unclear. 

12. This reviewer does not see the value of Figure one in the overall manuscript, or indeed the section on mapping. Besides clarifying the three observations periods, it is not really referenced or discussed further. This piece could be much reduced or deleted. 

13. Table One; the 9 family households were more likely to have mobile phones, radios, and solar electricity. These markers of wealth, especially in rural Ethiopia, suggest the households chosen were different than the average households in this area. What is the impact on the findings from this difference-please add this to the discussion and consider the ramifications, especially in terms of generalizability. 

14. In several places, the authors refer to how rare certain observations were (example-Laundry, line 260). Yet with only one day of observation , is it really so rare, or is the alleged scarcity a function of the few days observed. If laundry was done 2-3 times per week, which is rather a lot, it still is unlikely to be observed in the one observation day out of 7 days in a week. The authors need to temper their conclusions on the rarity of observations by acknowledging that the observation time itself was very limited. 

14. Table 3 and 4 are exceedingly difficult to interpret. Are these numbers of times observed> Were the same number of persons observed at each season? what should we consider as a reference? How are we to interpret 112 instances of handwashing only in the dry season? Some benchmark is needed, like a per person or per day or something. The tables state these are column percentages, but they appear to be row percentages within seasons? 

15. Please indicate the denominator for each person group, and the number not observed in each season. There is drop off over time, and this is not discussed. 

15. line 230-233. Are we to understand that water collection was only observed on 13 of the 27 days of observation? And that there were 32 successful trips-on these 13 says? That is almost 3 trips per day, with none on the other days? How many families contributed to the 13 days and the 32 trips? It is just hard to understand how clustered these data are, and how limited by the few days observed. 

16. in Table 2, there is almost a ten-liter difference, just in the dry season, between estimated water use and water use between 6:30 and 17:30 using the same method. Why was this so? Based on other statements, the authors feel most water use is in the morning, but in this case, it suggests virtually a third was used in the evenings. Does this mean then that for the dry season at least, the water consumption per person was a third higher, as it seems a third was used after 17:30? 

17. the authors state that water use for drinking was not collected because it was so modest. With between 5-8 persons per household, it is difficult to imagine that water use for drinking was not at least as great as that given to animals or to water the floor. Is the problem that water consumption could not be quantified for drinking? Or that water for drinking was often consumed off premises? Water for drinking and cooking are primary uses in my experience, admittedly in several other cultures, so it is difficult to reconcile the absence of this category in these data. 

Discussion

The authors need to tone down their assumptions of the generalizability of this study and acknowledge its limitations. For example, in line 314, the implication is these data are representative of rural Oromia, which clearly cannot be the case. In line 230, the statement is made “and three of the nine households did not collect water from any source in any season”. Again, clearly not the case -the three households DID collect water at some point, just were not observed collecting water on the day of observation. Please go through and review the manuscript and temper such statements.

Reviewer #2: This study explored a topic that is of high interest for the elimination of trachoma worldwide. There are very few studies involving the analysis of face washing practices in communities affected by trachoma that include direct observation of householders for several hours in different seasons of the year. This is a relevant study for countries affected by trachoma and to identify better interventions to tackle context-specific characteristics related to the endemicity of the disease. Congratulations to the authors for contributing with this study to the body of evidence on better practices for facial cleanliness.

Reviewer #3: Manuscript revision

Manuscript number: PNTD-D-21-01045 

Title: Seasonal variation in water use for hygiene in Oromia, Ethiopia, and its implications for trachoma control: an intensive observational study 

Short Title: Water use for hygiene and trachoma 

Keywords: Trachoma; water availability; hygiene, face washing; behaviour change.

In the DECS/MeSH data base I did not find: water availability, face washing and behaviour change. There are: water; face and behaviour. I would suggest the DECS/MeSH Terminology – trachoma; prevention & control; health risk behaviours or risk reduction behavior, health promotion or health education. 

General comments

The manuscript presents a study conducted in Ethiopia exploring seasonal variation in hygiene behaviours to inform the development of face washing interventions for trachoma control.

It is a very interesting paper that assessed hygiene habits in a community in the different seasons of the year. Shows how a different culture behave where water is scarce. It also presents suggestions for improving the F component of the SAFE strategy.

I would suggest that the numbers of the percentages should be standardized, one number or without a number after the dot throughout the text. 

At the abstract page 2 line 30, the hour is written “0630 and 2130” – the reader needs to guess that those numbers refer to hours. Only later in the text, that it was better explained.

At the table 2 line 218 – it is written “06:30 am” and in page 14 line 272, it is written “530pm and 930pm”. Please refer to the hours and minutes and use the abbreviations units officially accepted for the use with the International System of Units (SI): Hours – h; minutes – min. Standardize the use throughout the text. I would suggest writing 06:30h or 1h48min. 

I would suggest including in all the tables’ title the country name and the date.

PLOS authors have the option to publish the peer review history of their article (what does this mean?). If published, this will include your full peer review and any attached files.

Reviewer #1: No

Reviewer #2: No

Reviewer #3: No
---

## [Decision Letter · Decision Letter 1]

19 Dec 2021

Dear Dr Last,

Thank you very much for submitting your manuscript "Seasonal variation in water use for hygiene in Oromia, Ethiopia, and its implications for trachoma control: an intensive observational study" for consideration at PLOS Neglected Tropical Diseases. As with all papers reviewed by the journal, your manuscript was reviewed by members of the editorial board and by several independent reviewers. The reviewers appreciated the attention to an important topic. Based on the reviews, we are likely to accept this manuscript for publication, providing that you modify the manuscript according to the review recommendations. 

The manuscript addresses the issue of water use, hygiene practices, and endemic trachoma using a qualitative methodology. Generally, reviewers agreed that the revised version addressed most of the issues raised. However, one of the reviewers is still unconvinced that all problems were adequately assessed. I ask the authors to check whether any of the issues raised by Reviewer #1 can be addressed at this point. In particular, I feel it necessary to have specific answers and eventual changes in the manuscript regarding: (1) the definitions used for ocular and nasal discharge and (2) the Hawthorne effect.

Sincerely,

Guilherme L Werneck

Deputy Editor

Guilherme Werneck

Deputy Editor

The manuscript addresses the issue of water use, hygiene practices, and endemic trachoma using a qualitative methodology. Generally, reviewers agreed that the revised version addressed most of the issues raised. However, one of the reviewers is still unconvinced that all problems were adequately assessed. I ask the authors to check whether any of the issues raised by Reviewer #1 can be addressed at this point. In particular, I feel it necessary to have specific answers and eventual changes in the manuscript regarding: (1) the definitions used for ocular and nasal discharge and (2) the Hawthorne effect.

Reviewer's Responses to Questions

**Key Review Criteria Required for Acceptance?**

**Methods**

-Are the objectives of the study clearly articulated with a clear testable hypothesis stated?

-Is the study design appropriate to address the stated objectives?

-Is the population clearly described and appropriate for the hypothesis being tested?

-Is the sample size sufficient to ensure adequate power to address the hypothesis being tested?

-Were correct statistical analysis used to support conclusions?

-Are there concerns about ethical or regulatory requirements being met?

Reviewer #1: While I appreciate the responses to my concerns, there are still some major issues remaining. 

 The request for sample size determination has not been addressed. The response of selecting 9 households pragmatically and issues of feasibility are certainly real issues but cannot be used to justify a study where measurement error is clearly a possibility. IF significance testing is important, as it appears to be in comparing across seasons, then some determination of sample size to detect differences must have been made, or at least some determination of the power to detect differences given the existing sample size. 

The generalizability issue is still problematic. The kebeles were selected to match another study and the houses were chosen in order of a convenience sample of children undergoing a survey and who had trachoma. It would seem prudent to severely limit the generalizability of these findings, especially in section staring line 435 which discusses implications for interventions. 

In the methods section, please provide the definitions used for ocular and nasal discharge, and some indicator of agreement among observers.

Reviewer #2: Authors adjusted the methods section.

Reviewer #3: yes

**Results**

-Does the analysis presented match the analysis plan?

-Are the results clearly and completely presented?

-Are the figures (Tables, Images) of sufficient quality for clarity?

Reviewer #1: The response to concern that the trachoma survey may have influenced the family’s behavior if they knew the status of their children was reassuring, and the manuscript should include a statement to the effect that the families were not informed that their children had signs of trachoma as all were treated regardless.

 I am still a bit unclear on the rationale for why, if families made the connection between trachoma and hygiene, increased face washing would be more apparent on the second or third visit than the first. In our experience with observations, it is in fact the first visit, where the families do not know the observer or what they are observing that they try and be on best behavior. The response provided was survey was done on reported water availability and use after observations in the dry and in the rainy season (line 164) but washing did not increase in the rainy and harvest seasons. While the survey may have changed behaviors in subsequent seasons, it does not address the concern that first observations could be biased due to the Hawthorne effect. In any case, the fact that the first observations were completely confounded by season is real, and possible bias could exist and needs to be refuted by data that this is not a source of bias on seasonal data than disagreement that it is not so. Or discussed as a possibility in limitations. Of note, the limitation section has been expanded but still does not address the Hawthorne effect of first observations.

Reviewer #2: Results were also adjusted.

Reviewer #3: yes

**Conclusions**

-Are the conclusions supported by the data presented?

-Are the limitations of analysis clearly described?

-Do the authors discuss how these data can be helpful to advance our understanding of the topic under study?

-Is public health relevance addressed?

Reviewer #1: I remain unconvinced that there is real seasonality in the observations, given my concerns addressed above.

Reviewer #2: Conclusions are well written.

Reviewer #3: The conclusions are of public health relevance because identified some seasonal variation in hygiene behaviours and water consumption and suggested how better improve the facial cleanness of children in that setting to prevent trachoma.

**Editorial and Data Presentation Modifications?**

Reviewer #1: (No Response)

Reviewer #2: Accept

Reviewer #3: There are still some hours that are not standardized. Please correct in lines 28, 30, 31, 197, 300 and 322.

**Summary and General Comments**

Reviewer #1: (No Response)

Reviewer #2: The adjusted version looks very well. Authors included several clarifications.

Reviewer #3: The manuscript presents a study conducted in Ethiopia exploring seasonal variation in hygiene behaviours and water consumption in 9 households. The authors achieved their objectives to document facial hygiene practices and any seasonal variation that could be relevant for trachoma control.

I find very interesting this qualitative research to assess hygiene habits. The methods a section was well explained so can be reproduced by other researches in another settings. The results and conclusions were in line with the data collected. Despite the 9 households had more wealth, they showed a very low water consumption and barriers to face washing and use of soap. 

Congratulations

PLOS authors have the option to publish the peer review history of their article (what does this mean?). If published, this will include your full peer review and any attached files.

Reviewer #1: No

Reviewer #2: No

Reviewer #3: No

Figure Files:

Data Requirements:

Reproducibility:

References

---

## [Editor Report · Decision Letter 2]

18 Apr 2022

Dear Dr Last,

We are pleased to inform you that your manuscript 'Seasonal variation in water use for hygiene in Oromia, Ethiopia, and its implications for trachoma control: an intensive observational study' has been provisionally accepted for publication in PLOS Neglected Tropical Diseases.

Best regards,

Expedito J A Luna, MD

Guest Editor

Guilherme Werneck

Deputy Editor

The authors have appropriately addressed the issues raised by Reviewer #1.

---

## [Editor Report · Acceptance letter]

9 May 2022

Dear Dr Last,

We are delighted to inform you that your manuscript, "Seasonal variation in water use for hygiene in Oromia, Ethiopia, and its implications for trachoma control: an intensive observational study," has been formally accepted for publication in PLOS Neglected Tropical Diseases.

Best regards,

Shaden Kamhawi

co-Editor-in-Chief

Paul Brindley

co-Editor-in-Chief
